# Crystallization Induced Enhanced Emission in Two New Zn(II) and Cd(II) Supramolecular Coordination Complexes with the 1-(3,4-Dimethylphenyl)-5-Methyl-1*H*-1,2,3-Triazole-4-Carboxylate Ligand

**DOI:** 10.3390/polym12081756

**Published:** 2020-08-06

**Authors:** Pilar Narea, Jonathan Cisterna, Alejandro Cárdenas, Pilar Amo-Ochoa, Félix Zamora, Clàudia Climent, Pere Alemany, Sergio Conejeros, Jaime Llanos, Iván Brito

**Affiliations:** 1Departamento de Química, Facultad de Ciencias Básicas, Universidad de Antofagasta, Avda, Universidad de Antofagasta, Campus Coloso, Antofagasta 02800, Chile; pilar.narea@uantof.cl (P.N.); jonathan.cisterna@uantof.cl (J.C.); 2Departamento de Física, Facultad de Ciencias Básicas, Universidad de Antofagasta, Avda, Universidad de Antofagasta, Campus Coloso, Antofagasta 02800, Chile; Alejandro.cardenas@uantof.cl; 3Departamento de Química Inorgánica, Universidad Autónoma de Madrid, 28049 Madrid, Spain; pilar.amo@uam.es (P.A.-O.); felix.zamora@uam.es (F.Z.); 4Institute for Advanced Research Chemistry (IAdChem), Universidad Autónoma de Madrid, 28049 Madrid, Spain; 5Condensed Matter Physics Center (IFIMAC), Universidad Autónoma de Madrid, 28049 Madrid, Spain; 6Departamento de Física Teórica de la Materia Condensada, Universidad Autónoma de Madrid, E-28049 Madrid, Spain; claudiacliment@gmail.com; 7Departament de Ciència de Materials i Química Física and Institut de Química Teòrica i Computacional (IQTCUB), Universitat de Barcelona, 08028 Barcelona, Spain; 8Departamento de Química, Facultad de Ciencias, Universidad Católica del Norte, Avda, Angamos 0601, Antofagasta, Chile; sconejeros@ucn.cl (S.C.); jllanos@ucn.cl (J.L.)

**Keywords:** supramolecular coordination complexes, hydrogen bonded metal–organic frameworks, dihydrogen bonding-type, aggregation induced emission, multivariate properties

## Abstract

Two new *d*^10^ metal supramolecular metal–organic frameworks (SMOFs) with general formula [ML_2_(H_2_O)_2_]_n_ (M = Zn, Cd) have been synthetized using the sodium salt of the anionic 1-(3,4-dimethylphenyl)-5-methyl-1H-1,2,3-triazole-4-carboxylate ligand (Na^+^L^−^). Both SMOFs have been structurally characterized by single-crystal X-ray diffraction analysis and IR spectroscopy. The compounds are isostructural and form supramolecular aggregates via hydrogen bonds with the presence of less common dihydrogen bonds. Interestingly, they show ionic conductivity and porosity. The luminescent properties have been also studied by means of the excitation and emission spectra. Periodic DFT and molecular TD-DFT calculations have been used to unravel the emergence of luminescence in the otherwise non-emitting 1-(3,4-dimethylphenyl)-5-methyl-1H-1,2,3-triazole-4-carboxylate ligand once incorporated in the SMOFs. Our results also illustrate the importance of considering the dielectric environment in the crystal when performing excited state calculations for isolated fragments to capture the correct electronic character of the low-lying states, a practice which is not commonly adopted in the community.

## 1. Introduction

One of the most active areas in materials chemistry in recent years involves the quest for novel organic solid-state luminescent materials for the development of organic light-emitting diodes (OLEDs) [1,2,3], bioimaging agents [4,5,6], chemosensors, detectors of microenvironmental changes [7,8,9,10], or dynamic functional materials [11,12,13]. Unfortunately, the most efficient organic luminogens that are known contain planar π-conjugated aromatic rings, for which it is well established that formation of aggregates might have a large influence on the intensity of the luminescent emission, in most of the cases leading to its quenching, a phenomenon often described in the literature as ‘aggregation caused quenching’ (ACQ).

Several strategies have been developed to overcome this inconvenience which severely limits the application in solid-state devices of many materials which otherwise have superb luminescent properties in solution. Most of these efforts are basically aimed towards controlling the aggregation process to prevent excimer formation [14,15,16,17] that is thought to be at the origin of ACQ in many cases [18,19]. A breakthrough in the field occurred in 2001 when Tang and co-workers observed that 1-Methyl 1,2,3,4,5-pentaphenylsilole, a poor luminogen which was hardly emissive in common organic solvents, became surprisingly highly emissive upon aggregation [20]. The term aggregation induced emission (AIE), coined to describe this phenomenon [21,22,23,24], is now widespread in the field of luminescent materials [25,26]. Besides the general AIE term which applies to all kinds of aggregated states, in the particular case where the emission is induced by crystallization as in the present case, the terms CIE (crystallization induced emission) and CIEE (crystallization induced emission enhancement) have also been used in the literature [27]. Despite the success of new purely organic AIE-active molecules, in the years following Tang’s group discovery it became obvious that the introduction of heavy atoms—especially transition metals—in these AIE-active molecules was necessary for the development of new optoelectronic devices since efficient purely organic phosphorescent molecules are very scarce [27,28,29,30,31].

It is known that the efficient organization of molecules via supramolecular interactions results in the restriction of the intramolecular motion (RIM) and the blockage of the non-radiative deactivation pathways giving rise to the activation of the AIE process [12,13,14]. Different approaches have been used to study the effect of intramolecular motion in the AIE mechanism as, for instance, computational modelling or, from an experimental point of view, the control of the solution’s viscosity [15,16,17,18,19]. It is, however, still hard to precisely determine and control how aggregation actually limits intramolecular motion giving rise to AIE. Besides restriction of the intramolecular motion, inclusion of heavy metal atoms, leading to intensity borrowing of the lowest triplet state from bright singlet states, combined with the intrinsic photophysical properties of AIE-active π-conjugated ligands has been a much sought after strategy for obtaining new luminescent functional materials.

The use of supramolecular interactions such as hydrogen-bonds can form organized materials based on the assembly of molecules as building blocks. The selection of the molecular components and the interactions between them define the structure and the properties of the final material. Hydrogen bonds are also frequently used to connect organic or metal–organic molecules giving rise to well-organized frameworks that are structurally flexible and dynamic because of the nature of these bonds. Thus, the connection of coordination compounds by H-bonds gives rise to supramolecular coordination complexes (SCCs) or supramolecular metal–organic frameworks (SMOFs) [32] which can be designed with a large variety of structures and properties going from rigid to flexible porous materials. Moreover, SCCSs can incorporate emissive building blocks leading to interesting luminescent materials [33]. In this sense, the systematic synthesis of SCCSs using appropriate organic ligands and metal centers has been a fruitful approach to obtain new luminescent materials, especially with systems containing *d*^10^ metal centers [34,35].

N-donor linkers have been widely adopted for the assembly of tunable coordination networks. From the point of view of coordination chemistry, this nitrogen rich heterocycle exhibits abundant coordination modes [36,37]. Moreover, nitrogen-containing heterocycles are susceptible to form hydrogen bonds, a feature that may be beneficial for the self-assembly of a supramolecular structure in the solid state. Taking these facts into account, multidentate ligands based on 1,2,3-triazole may be considered as versatile building blocks for new functional materials with emerging properties that can be fine-tuned just by a judicious choice of the substituents on the triazole rings. The incorporation of carboxylate groups enhances the coordination capability, enabling the growth of the coordination network in the coordination compounds [38,39,40]. Interactions between this type of ligands and metal centers could give information about their synergistic capability, on the antenna effect, charge transfer transitions, or π-system extension that have been studied previously in other systems [41,42,43,44,45,46,47,48].

In this communication, we report on the synthesis and characterization of a new class of Zn(II) and Cd(II) SMOFs including the 1-(3,4-dimethylphenyl)-5-methyl-1*H*-1,2,3-triazole-4-carboxylate anion as a ligand. The structure of two new SMOFs has been determined by single-crystal X-ray diffraction and the resulting supramolecular architecture rationalized using the semi-empirical Hirshfeld method to analyze the topologies of their energy framework. This new class of supramolecular metal–organic frameworks presents interesting dihydrogen-bond induced interactions, an unusual type of interactions for which, to the best of our knowledge, only a few examples have been reported and studied previously [49,50]. The two newly synthesized SMOFs show thermal and luminescent properties that are studied by solid state fluorescence spectroscopy. Interestingly, while solutions of the sodium salt of 1-(3,4-dimethylphenyl)-5-methyl-1*H*-1,2,3-triazole-4-carboxylate do not show any detectable emission in the UV–visible region, coordination with the Zn(II) and Cd(II) cations induces a luminescent response of the ligand itself. In order to understand the origin of this crystallization induced enhancement of emission (CIEE) in these SMOFs we also carried out periodic DFT and molecular TD-DFT based calculations to unravel the nature of the emitting states in the solid state and the role of the metals in the observed AIE effect.

## 2. Materials and Methods

### 2.1. Synthesis

Except for the solvents, which were purified and dried according to the method described by Armarego et. al. [51], the rest of chemicals were A.R. grade and used without further purification (Sigma-Aldrich, St. Louis, MO, United States). The syntheses of the ester precursor and the sodium salt of the carboxylate ligands were carried according to the procedure previously described by Brito et al. and reported elsewhere [38]. On the other hand, the 1-bis-ethyl 1-(3,4-dimethylphenyl)-5-methyl-1*H*-1,2,3-triazol-4-carboxylate diacquo Zn(II) or Cd(II) complexes (compounds 1 and 2, respectively), were obtained by double layer solution diffusion in crystallization at room temperature using a test tube (15 cm length by 1.5 cm diameter) between 50 mg (0.197 mmol, 2 eq.) of the sodium salt of the carboxylate ligand, in 5 mL of water, and 29.3 mg (0.0985 mmol, 1 eq.) of Zn(NO_3_)_2_·6H_2_O for compound 1 or 30.4 mg of Cd(NO_3_)_2_·4H_2_O (0.0985 mmol, 1 eq.) for compound 2, in 5 mL of *n*-BuOH, respectively. The crystals formed at the interphase were filtered off, washed with water, and dried.
(1)Colorless parallelepipeds, 34.3 mg (yield 31.0%, based on C_12_H_12_N_3_NaO_2_). FT-IR (KBr pellet, cm^−1^): 3417(vs) ν(O–H), 3169(w), 3043(w) ν(C_sp_^2^–H), 2949(m), 2924(m) and 2854(w) ν(C_sp_^3^–H), 1614(vs) ν(C=O and/or C=C), 1576(s) ν(C–N), 1400(vs) ν(N=N and/or -CH_3_), 1290(m) ν(C–O), 1248(w) ν(C_sp_^2^–N), 532 (vw) ν(Zn–O).(2)Colorless blocks, 47.7 mg (yield 40.0%, based on C_12_H_12_N_3_NaO_2_). FT-IR (KBr pellet, cm^−1^): 3412(vs) ν(O–H), 3198(w) ν(C_sp_^2^–H), 2945(m), 2924(m) and 2860(w) ν(C_sp_^3^–H), 1605(vs) ν(C=O and/or C=C), 1572(s) ν(C–N), 1402(vs) ν(N=N and/or -CH_3_), 1286(m) ν(C–O), 1246(w) ν(C_sp_^2^–N), 808 (vw) ν(Cd–O).

### 2.2. Characterization

FT-IR spectra in the range 400–4000 cm^−1^ were recorded on a Nicolet Avatar 300 spectrometer using KBr pellets (Thermo Scientific, Waltman, MA, USA). Excitation (PLE) and emission (PL) spectra were measured at room temperature using a Jasco FP-8500 spectrofluoremeter with a 150 W xenon lamp as the excitation source (JASCO Co.; Kyoto, Japan). The emission spectra were obtained by exciting the samples at 450 nm. Optical band gaps were determined from the diffuse reflectance spectra, which were measured with a Perkin-Elmer Lambda 20 UV–vis spectrophotometer equipped with a Labsphere RSA-PE-20 diffuse reflectance and transmittance accessory in the range of 200–600 nm (6.2–2.1 eV) (Perkin-Elmer, Akron, OH, USA).

### 2.3. X-ray Structural Determination

Diffraction data at 293–295 K were obtained for suitable crystals of compounds 1 and 2 on a Bruker D8 Venture diffractometer equipped with a bidimensional CMOS Photon 100 detector, using graphite monochromated Cu-Kα (*λ* = 1.54178 Å) radiation (Bruker Co.; Billerica, MA, USA). The diffraction frames were integrated using the APEX3 package [52] and were corrected for absorptions with SADABS [53]. The structures of 1 and 2 were solved by intrinsic phasing [54] using the *OLEX2* software [55] and refined with full-matrix least-square methods based on *F*^2^ (*SHELXL*) [56]. For both compounds, non-hydrogen atoms were refined with anisotropic displacement parameters. All hydrogen atoms were included in their calculated positions, assigned fixed isotropic thermal parameters, and constrained to ride on their parent atoms. In both structures, a solvent mask procedure was used to omit two disordered water molecules (compound 1), and residual electron density over crystal voids (compound 2), using the *OLEX2* software [55]. A summary of the details about crystal data, collection parameters and refinement are documented in Table 1, while additional crystallographic details are included in the CIF files. ORTEP views were drawn using the *OLEX2* software [55]. The crystallographic data and details of the structure refinements are summarized in Table 1. CCDC no. 2005759 (1) and 2005760 (2) contain the supplementary crystallographic data for this paper. These data can be obtained free of charge from the Cambridge Crystallographic Data Center at http://www.ccdc.cam.ac.uk/data_request/cif.

### 2.4. Computational Details

Hirshfeld surface analysis [57] was performed using CrystalExplorer 17.5 [58] and comprised d_norm_ surface plots and 2D (two-dimensional) fingerprint plots [59]. The electrostatic potentials were mapped on the Hirshfeld surfaces using the 3-21G(d, p) basis set at the level of Hartree–Fock theory over a range of ±0.002 au. The crystallographic information file (CIF) was used as input for the analysis. For the generation of fingerprint plots, the bond lengths of hydrogen atoms involved in interactions were normalized to standard neutron diffraction values (C-H = 1.083 Å, N-H = 1.009 Å, O-H = 0.983 Å). The intermolecular energies of the molecular pairs in the crystal packing were calculated, at the HF/3-21G (d,p) level of theory, in a cluster of radius 3.8 Å around the molecule [58], and their respective energy framework diagram are shown in Appendix A. The solvent masking procedure as implemented in *OLEX2* was used to remove the electronic contribution of solvent molecules and residual electron density from the refinement. This procedure does not affect the Hirshfeld and energy framework model calculations. All the surface, energy frameworks, and plots of the Hirshfeld surface were drawn using the CrystalExplorer software [58].

Periodic ab initio density functional theory (DFT) calculations were performed using the CRYSTAL17 code [60,61,62] to evaluate the electronic properties of both compounds in the solid state. The electronic structure was calculated using the hybrid B3LYP functional [63]. All-electron Gaussian-type basis sets were used for O, N, C, H [64,65], and effective-core pseudopotentials (ECP) with a valence basis set for Zn and Cd [66,67]. For the calculation of the Coulomb and exchange integrals, tolerance factors of 7, 7, 7, 9, and 30 were used and the convergence criterion for the electronic energy was set at 10^−7^ a.u. [62]. A mesh of 8 × 8 × 8 k-points in the irreducible Brillouin zone, chosen according to the Monkhorst–Pack scheme [68], was used for reciprocal space integrations. The convergence of the total energy with the grid size was checked. All periodic calculations presented were performed using the experimental geometries, keeping the space group symmetries fixed.

We also performed DFT calculations for the isolated ligand anion L^−^ and the [M(L)_2_·(H_2_O)_2_] (M = Zn, Cd) fragments in vacuum and embedded in a continuous dielectric with different values for the dielectric constant. These calculations were performed using the Gaussian [69] software and the integral equation formalism variant of the polarizable continuum model (IEFPCM) [70,71]. Excited state energies were obtained from time dependent DFT (TD-DFT) calculations using the CAM-B3LYP functional [72] in combination with a standard 6-31+G(d) basis set. The choice of a long-range corrected functional was motivated by the presence of charge transfer transitions. Note that although this type of functional usually yields transition energies larger than the experimental ones, they provide an accurate description of the nature of the states, which was the main concern in this work.

## 3. Results and Discussion

### 3.1. Synthesis and Characterization

Neutral mononuclear complexes with general formula [M(L)_2_·(H_2_O)_2_] were obtained according to the procedure previously reported (see Figure 1) [38]. Slow diffusion of the reactants in immiscible liquids (*n*-BuOH/H_2_O) resulted in the formation at the interphase of single crystals suitable for XRD analysis. Nitrate salts were used as a metal source. In general, both compounds were obtained with a moderate yield (30–40%). Moreover, the two products are insensitive to moisture and exhibit a large thermal stability until total decomposition. Both [M(L)_2_(H_2_O)_2_] complexes have a poor solubility in common organic solvents as well as in water. The composition of the two new compounds was deduced from FT-IR spectroscopy over crystalline samples. Additionally, the crystal and molecular structures of **1** and **2** were determined by single crystal X-ray diffraction analysis (see Section 3.3).

### 3.2. FT-IR Spectroscopy

FT-IR spectra of **1** and **2** exhibit a similar band pattern and their respective spectra are depicted in Appendix A. Both compounds show a broad band at ca. 3400 cm^−1^ attributed to the O–H stretching vibration of the aquo axial ligands involved in hydrogen bond interactions of the supramolecular aggregates (see Section 3.3 for more details). The bands in the 1600–1630 cm^−1^ region were assigned to ν(C=C) and ν(C=O) of the stretching vibration modes of the aromatic ring and the carboxylate fragment in the ligand, respectively. In the case of ν(C=O) vibration modes, these are down-shifted, due to chelating effect to the metal center, being indicative of a substantial delocalization of the electron density throughout the entire π-conjugated system that may, a priori, enhance their emissive properties (see Section 3.5 for more details). However, in the case of ν(C=C) vibration no significant shifts were observed. This can be explained considering that the dihedral angles between the 3,4-phenylene and 1,2,3-triazole ring are larger, leading to a loss of electronic communication between both fragments (see Section 3.3 for more details). Both compounds also show a sharp band around 800 cm^−1^, which can be assigned to the 1,3,4-trisubstituted phenylene ring.

### 3.3. X-ray Crystallographic Studies

Single-crystal X-ray diffraction analysis reveals that **1** and **2** are isotypic, crystallizing in the centrosymmetric monoclinic space group *P2_1_/c* with 2 molecules per unit cell. The basic structural unit in both cases is a mononuclear Zn(II) or Cd(II) complex with two 1-(3,4-dimethylphenyl)-5-methyl-1*H*-1,2,3-triazole-4-carboxylate ligands (L^−^) and two water molecules. In both compounds the asymmetric unit contains one half of Zn(II) or Cd(II) cations, one L^−^ anion, and one coordination water molecule, respectively. Metal atoms are coordinated to two oxygen and two nitrogen atoms coming from two distinct L^−^ ligands (O1, O1^i^; N1, N1^i^) and two O atoms belonging to two coordination water molecules (O2, O2^i^). Each metal is located on an inversion center with a severally distorted *trans*-N_2_O_4_ octahedral environment with the N and O donor atoms from the ligands in the equatorial plane and two O axial ligands coming from the water molecules (see Figure 2). In both compounds the carboxyl group is mono-coordinated, generating a chelate with an N-donor atom from the triazole ring. The Zn−O bond lengths in **1** range from 2.070(3) to 2.127(3) Å and the Zn−N bond length is 2.147(3) Å. On the other hand, for compound **2**, the Cd−O bond lengths range from 2.278(4) to 2.321(5) Å and the Cd−N bond length is 2.311(6) Å.

The dihedral angles between the planes of the (3,4-dimethyl)-phenyl fragment and the triazole ring are 66.6(2) and 65.7(4)°, in **1** and **2**, respectively, while the triazole and metallacycle rings are practically coplanar in the two compounds with dihedral angles of 4.10(13) and 3.89(19)°, respectively. All bond distances and angles are in the range previously reported for related compounds [73,74,75,76]. Selected bond distances and angles are summarized for both compounds in Appendix A.

Figure 2 shows the basic structural unit in the crystals of **1** and **2** that consists of ML_2_(H_2_O)_2_ complexes. These discrete fragments arrange in the crystals forming H-bond induced one-dimensional [M(LOH_2_)_2_]_n_ linear chains running parallel to the [001] axis (Figure 3). There are two H-bonds between each pair of ML_2_(H_2_O)_2_ complexes, where the O2 atoms on the water molecules act as donors and the uncoordinated O3 atoms of the carboxylate group as acceptors. These O(2)–H2A···O(3)^i^–C paths form $R_{2}^{2}$(12) centrosymmetric rings (labeled as A in Figure 3). The [ML_2_(H_2_O)_2_]_∞_ chains are linked to neighboring chains by two further types of non-covalent interactions: (a) a weak dihydrogen-type interaction O(2) –H2B ···H6^ii^ – C that forms $R_{2}^{2}$(16) centrosymmetric rings (labeled B in Figure 3), and (b) a mixture of O(2) –H2A···O(3)^i^ – C and O(2) –H2B ···H6^ii^ H-bonds that form $R_{2}^{2}$(24) centrosymmetric rings labeled C in (Figure 3) [77] (for symmetry codes “^i^” and “^ii^” see Appendix A). These three types of rings alternate in a …ABCABC… pattern to form the two-dimensional supramolecular aggregates shown in Figure 3. The list of distances and angles involving hydrogen-bond interactions described above can be found in Appendix A. Dihydrogen-type bond interactions, which are similar in all respects to hydrogen-bonds, fall in the class of van der Waals interactions [50]. They are less common than conventional H-bonds but have already been studied both experimentally and theoretically [49,50,78]. According to their geometrical preferences, this type of interaction can be considered as a hydrogen-bond involving a further hydrogen atom with pronounced hydridic character as the acceptor [49]. The crystal structure shows very weak H···H interactions in the direction parallel to plane ‘A’ that result in a three-dimensional multiple strand, layer by layer structure.

### 3.4. Hirshfeld Surface Analysis

Hirshfeld surface analysis [59] was conducted to verify the contributions of the different intermolecular interactions to the formation of the supramolecular structure (see experimental section for more details), and these are shown in Table 2 This analysis was used to investigate the presence of hydrogen-bonds and other weak intermolecular interactions in the crystal structure). The plots of d_norm_ on the Hirshfeld surface for a single [ML_2_(H_2_O)_2_] fragment (Figure 4) confirm the presence of the non-covalent interactions in the planes described above (see Section 2.4) with the red spots indicating the position of the main, H-bonding, interactions.

Compounds **1** and **2** have nearly identical d_norm_ surfaces because both compounds are isostructural and isomorphous. The fingerprint plots display a symmetrical shape because the intermolecular contact between one unit (promolecule) and their environment (procrystal) are reciprocal and symmetrical. On the other hand, to visualize and quantify the similarities and differences in intermolecular contacts across the crystal structure, the Hirshfeld surface analysis was made with complementary descriptors such as the shape index surface (Figure 5).

The weak intermolecular interactions present in **1** and **2** are mainly of the H···O, H···N, H···C and H···H types. The reciprocal contacts appear in the fingerprint plot (Figure 4) as a sharp needles for H···O, with *d*_e_ + *d*_i_
~ 2.7 Å. H···N appear as two double diffuse wings, pointing at a distance greater than the van der Waals radii of N and H atoms, with *d*_e_ + *d*_i_
~ 2.9 Å (d_i_ + d_e_ > 2.75 Å) [79], and hence, with no significant contribution towards the crystal packing. The H···C interactions show up as symmetrical thick wings with *d*_e_ + *d*_i_
~ 3.0 Å, mainly as a consequence of weak H···π interactions. The interatomic H···H contacts have a major contribution in the fingerprint plot appearing as a symmetrical wide stain with *d*_e_ + *d*_i_
~ 2.4 Å [79], denoting therefore the presence of short H···H contacts participating in a significant way in the molecular packing leading to the crystal structure stabilization.

The shape index in Figure 5 allows us to determine the presence of H···π interactions, where the yellow–orange spots show the surface subsidence due to the proximity of the neighboring moieties and the blue green spots show the reciprocal contacts of the moieties that generate the subsidences. In this case, their respective counterparts, where the interaction between hydrogen and dihydrogen bonds is established when generating the crystal structure, are shown in Figure 5. However, the sum of VdW radii for C and H atoms interacting in Figure 5 is slightly larger than their theoretical value (3.17 vs. 2.9 Å). Thus, this type of interaction can be considered very weak. As for the d_norm_ surface, compounds **1** and **2** show nearly identical shape index surfaces.

### 3.5. Electronic Emission Spectra

The luminescent properties of **1** (φ = 5.67%) and **2** (φ = 6.08%) were evaluated and compared to those of the sodium salt of the ligand. All spectra were recorded in the solid state. Emission spectra are shown in Figure 6. Surprisingly, the ligand is barely emissive in solution, while showing similar spectra in the two new complexes in the same conditions. The sodium salt of the ligand is also emitting in the solid state, although as shown in Figure 6, the intensity of this emission is lower than for the two new SMOFs. It is possible to observe a small redshift of about 9 nm in both metal complexes compared with the ligand sodium salt. This small difference suggests that the metal centers are not involved in the excitation process, which should be attributed to intra-ligand transitions. However, this interpretation is not straightforward because if emission is only due to the ligand, it is not straightforward to explain why, these same ligands do not emit when isolated in a dilute solution (see Appendix A for more details).

### 3.6. Computational Studies

In order to shed some light into the mechanism by which metal complexation and crystallization lead to a sizable emission intensity in these two new materials, we have undertaken quantum chemical calculations. Thus, periodic hybrid DFT calculations were performed for **1** and **2** to understand their electronic structure and the contributions of weak interactions in the solid state on the nature of the states at the top of the valence band and bottom of the conduction band, respectively.

The B3LYP calculated band gaps are 5.5 and 5.2 eV for compounds **1** and **2**, respectively, somewhat larger than the corresponding experimental values of 4.35 and 4.18 eV obtained from the corresponding UV–vis diffuse reflectance spectra shown in Appendix A. The calculated values are in fair agreement with the experimental ones, indicating that the gap is somewhat smaller in the Cd compound.

We note for both cases the presence of almost flat bands near the Fermi level, confirming that these compounds behave as molecular solids. Examining the calculated density of states (DOS) in the region around the band gap for **1** and **2** (Figure 7 and Appendix A), we observe the absence of any significant metal contribution either at the top of the valence band or at the bottom conduction band. We can thus conclude that these states, which are mostly responsible of the luminescent emission, predominantly arise from ligand states. As already anticipated in the previous section from the comparison of the two emission spectra and that for the sodium salt of the ligand, states associated with the transition metal (3*d* or 4*d*) do not have any significant participation in the emission observed for solid samples of the two new compounds. In addition, the calculated DOS shows that the main contribution at the top of the valence band comes from the 2*p* states of the oxygen atoms in the carboxylate group, with a considerable participation of carbon 2*p* states. On the other hand, the main contribution at the bottom of conduction band arises from the 2*p* orbitals of the three nitrogen atoms located in the triazole ring of the ligand, pointing towards an intraligand charge transfer transition (ILCT) nature for the observed emission.

Additionally, in **1**, the Mulliken overlap populations involving the central Zn(II) ion and all six atoms (4 O and 2N) in its first coordination sphere indicate strong bonding interactions between Zn(II) and the two O atoms on the carboxylate fragment (0.137 e^−^) and between Zn(II) and the N atom on the triazole ring (0.52 e^−^). In contrast, the interaction between Zn(II) and the O atoms from the water molecule has a much smaller value (0.049 e^−^), pointing towards a much weaker interaction. The weak interaction between water molecules and the ligands is between the H atoms from water molecule and the O atoms on the carboxylate group (0.041 e^−^). On the other hand, there are no significant bonding interactions between ligands, since the overlap population values are very small (0.013–0.002 e^−^), pointing towards weak van der Waals interactions to hold the 3D crystal structure together as already anticipated from the Hirshfeld surface analysis above. Similar insights are obtained for **2**.

The comparison between the experimental and the computational data for the solid state samples indicate that luminescence has a common origin in both compounds and that it is related to intraligand charge transfer transitions (ILCT) while the metal atoms do not participate at all, besides a small indirect effect of shifting the involved states upon coordination. However, these results do not answer the crucial question: if the luminescence in these compounds has only its origin in the ligands, then why do we not observe it already in aqueous solutions of the sodium salt? In other words, what is the mechanism by which emission is induced in these compounds upon crystallization? Unfortunately, both **1** and **2** are practically insoluble in water or other organic solvents, so that it is not experimentally feasible to determine if luminescence is already present for isolated ML_2_(H_2_O)_2_ complexes or if it is a property that appears only upon ordered aggregation of these basic units to form the solid-state structure. In order to shed some light on these questions, we have undertaken a more detailed analysis of the nature of the excited states in these compounds using time dependent DFT (TD-DFT) calculations.

For this purpose, we computed the situation of a single isolated ligand and afterwards an isolated ML_2_(H_2_O)_2_ fragment and then compared these results with those obtained for the whole crystals. The results of TD-DFT calculations for the ten lowest excited singlet states of the isolated ligand L^−^ with the geometry taken from the experimental X-ray structure of **1** are shown in Figure 8. In these calculations we have considered the anionic ligand both in vacuum and in a continuum dielectric with different dielectric constants, ranging from a weakly polar case (dichloromethane, dcm, with ε_r_ ~8) to a strongly polar case (water with ε_r_ ~ 80) and an additional intermediate situation (acetonitrile, acn, with ε_r_ ~ 35). As expected for ionic species, transition energies calculated in vacuum are quite different from those calculated in a dielectric medium, with values between 1 and 2 eV lower than those observed for the vacuum case. The effect of the polarity of the environment is however very small and transition energies are practically the same for any of the three solvent models considered. A look at Figure 8b reveals that while the trends observed for transition energies in vacuum or in a solvent are quite similar, the behavior of the oscillator strengths between calculations in vacuum and those in a dielectric medium are radically different. While we do not get any appreciable oscillator strength for transitions to any of the 10 lowest excited singlet states in vacuum, we see that this is not the case for calculations considering a dielectric medium. Note, however, that all 10 excited states have quite modest oscillator strengths, especially the lowest lying ones, with S_5_ being the first one that has a non-negligible intensity. Since in our study we are considering emission spectra, we arrive to the conclusion that, in good agreement with the experimental data, the isolated ligand should be a quite bad fluorescence emitter because there are no low-lying singlet states that could lead to an appreciable emission intensity. To discard the possibility of a strong structural rearrangement upon coordination to the metal that would dramatically change the photophysical properties of the ligand, we have optimized the ground state geometry for the isolated anionic ligand both in vacuum and in water. Although there is a non-negligible change in the structure concerning the dihedral angle between the triazole and the phenyl rings (~70° in **1**, ~60° in water, and ~40° in vacuum), these changes apparently have no important consequences for the photophysical properties of the low energy excited states of the ligand (see Appendix A). In any case, we find negligible or quite low oscillator strengths for the lowest lying singlet states, and hence, we can safely deduce that the isolated ligand should not emit strongly in solution, irrespective of the solvent and the considered geometry.

To further gain some understanding on the origin of the emission observed for **1** and **2**, we have evaluated the nature of these low energy transitions for the isolated ligand (see Appendix A for the frontier molecular orbitals of the ligand). If we consider the calculations in vacuum, transitions to the five lowest singlet states are basically COO^−^ lone pairs → π* (phenyl, triazole, COO^−^) transitions. Since there is no overlapping between the lone pairs on the carboxylate fragment and the different π-orbitals, the computed oscillator strengths in the vacuum are practically zero. When considering a continuous dielectric medium, the transition to S_1_ is like in the case of vacuum, a COO^−^ lone pairs → π* transition, but transitions to S_2_ up to S_5_ states include also some π → π* character from the phenyl and triazole units. These states have, therefore, more localized π, π* character as well as charge transfer of π, π* type, where the orbitals do effectively overlap. This is the reason why these states have a larger oscillator strength in a dielectric medium. These conclusions are the same, irrespective of the solvent, with changes on the dielectric constant leading only to minor changes in the weights of the different orbitals in the transition amplitudes. Note that in all cases, due to the large amount of lone pairs and π orbitals of the different functional groups of the ligand, transitions are not just simple HOMO → LUMO excitations, with a complex mixture of orbital-to-orbital contributions in each case (see Appendix A for the main contributions to the S_5_).

With regard to the non-emissive nature of the anionic ligand in solution, we carried out further calculations and optimized the low-lying excited states of the anionic ligand in vacuum and in water. In both situations, we found that the most stable excited state minima correspond to a COO^−^ lone pairs → π* (phenyl, triazole, COO^−^) transition to the ground state with null oscillator strength, providing further support to the experimental observations. The torsion angle between the phenyl and triazole units decreases in this S_1_ minimum with respect to the ground state minimum, from ~40° to ~20° in the vacuum calculation and from ~60° to ~50° in water. We would like to note that a π,π* excited state minimum delocalized across the phenyl and triazole units with significant oscillator strength was found when considering a dielectric environment. However, such a minimum was found to be more than 0.3 eV above the COO^−^ lone pairs → π* (phenyl, triazole, COO^−^) minimum, with Kasha’s rule precluding emission from this higher-lying excited state.

TD-DFT transition energies and oscillator strengths for the 10 lowest singlet excited states for the ZnL_2_(H_2_O)_2_ fragment in compound **1** (considering the crystal geometry) are shown in Figure 9.

Just as in the calculations for the isolated ligand, for the Zn complex, moving from DCM (ε_r_ ~ 8) to water (ε_r_ ~ 80) has practically no effect on the energies and the nature of the excited states. However, the inclusion of a dielectric medium has an important effect compared to the vacuum case. Since the vacuum is not a sensible model for the complex in the crystal environment, although we do not exactly know the precise dielectric constant for the solid, we will limit our discussion to the results obtained in DCM, recalling that results for dielectric constants between 8 and 80 are very similar.

The first important thing to notice in the calculations for **1** is that the HOMO in the complex does no longer correspond to lone pair orbitals of the carboxylate group as for the isolated ligand (Figure 10). As found already in the DFT calculations for the whole crystal (Figure 7), the Zn orbitals have a negligible participation in the frontier orbitals. What is more important, coordination of the carboxylate groups with the metal and hydrogen bond interactions within the 2D layers stabilize the COO^−^ lone pair orbitals, which are no longer dominant in the nature of the low-energy excitations. The HOMO (Figure 10) in this case is a π-type orbital of the phenyl rings of the ligand. Due to the centrosymmetrical structure of the complex with two symmetry equivalent ligands and the metal atom located on the inversion center, the HOMO corresponds to the out-of-phase combination of the same phenyl π-type orbital of the two ligands, while the HOMO-1 orbital—very close in energy to the HOMO—corresponds to the in-phase combination of the same π-type orbitals. In an analogous way, the LUMO and LUMO+1 for the complex correspond to the in- and out-of-phase combinations of a triazole centered π-type orbital (Figure 10). Note that the ILCT character of the transition was well captured by our previously discussed solid-state calculations, that predicted a charge transfer from the phenyl ring to the pyrazole ring in each ligand constituting the complex. Note that in the density of states shown in Figure 7, besides the O contribution, there is a non-negligible C contribution arising from the phenyl rings at the top of the valence band.

Analysis of the TD-DFT calculations for the complex shows that the four lowest excited singlet states, with very similar energies among them, have all zero or very small oscillator strengths. Note that, due to the centrosymmetrical nature of the complex, all ligand centered transitions appear by pairs, with practically degenerate transition energies, where the lowest transition in each pair has a finite oscillator strength while the upper transition is forbidden. This can be readily understood from Kasha’s classical model for 1D aggregates where the excitonic couplings in the centrosymmetric complex emulate J-aggregates, speaking in terms of the transition intensities [80]. The low-lying bright excited state of the complex is S_5_ which corresponds to the sum of the charge transfer π(phenyl) → π*(triazole) transition on each ligand and it is therefore very bright. The S_6_ state, with zero oscillator strength and practically the same transition energy as S_5_, corresponds to the subtraction of these transitions. Since from Figure 10 it is evident that the metal orbitals do not significantly participate in the molecular orbitals involved in the S_0_ to S_5_ transition, it is clear that the same mechanism is responsible for the emission in **2**, explaining also the strong similarities between the two emission spectra shown in Figure 6. Although it would be necessary to have a detailed crystal structure of the sodium salt of the ligand to understand the origin for the emission in this case, the fact that the emission spectrum is quite similar to that found in **1** and **2**, although with lower intensity, it may be speculated that a similar mechanism is taking place in this case. Interaction of the carboxylate groups on the ligand with the Na^+^ cations in the crystal would lead to a stabilization of COO^−^ lone pairs, leading to an HOMO dominated by π(phenyl) orbitals that would result in a π(phenyl) → π*(triazole) nature for the observed transitions.

Note that, by coincidence, the bright low-lying state of the anionic ligand in the crystal is also the S_5_ state. However, the bright state of 1 does not simply correspond to the combination of each of the S_5_ states on each individual ligand (Appendix A), because the ligand by itself does not present this pure π(phenyl) → π*(triazole) transition, which only appears upon complexation with the metal atom. In this case, we find that the observed emission is not strictly speaking induced by the aggregation of the luminophores—i.e., the ligands—but by formation of the SMOFs, where the coordination of the carboxylate ligands to the central metal atom is essential. The role played by the metal atom in the emission process is, however, indirect since no appreciable contribution of metal centered orbitals can be found in the frontier orbitals involved in the low energy transitions. The metal atoms provide, on one hand, a way of stabilizing the COO^−^ lone pair orbitals, which lie in the region of the highest occupied orbitals in the bare ligand, changing in this way completely the nature of the low energy transitions of the ligand once it is incorporated into **1** or **2**. On the other hand, it is important to remark that the oscillator strength calculated for the lowest bright state of the complex, which is much larger than for any low energy transition in the isolated ligand, is basically due to the centrosymmetric arrangement of the two ligands around the metal center which gives rise to a pair of excited states, one bright and the other one dark, for each transition in the individual ligands forming the complex.

## 4. Conclusions

We reported the synthesis of two new *d*^10^ metal supramolecular metal–organic frameworks (SMOFs) of general formula [ML_2_(H_2_O)_2_]_n_ where M = Zn or Cd and L corresponds to the anionic 1-(3,4-dimethylphenyl)-5-methyl-1*H*-1,2,3-triazole-4-carboxylate ligand. The discrete centrosymmetric ML_2_(H_2_O)_2_ units arrange into linear chains through a network of hydrogen bonds between the carboxylate groups of the ligand and the coordinated H_2_O molecules. Interestingly, these chains form 2D layers held together by quite unusual dihydrogen bonds.

The most striking feature of these new compounds is their crystallization induced emission. While aqueous solutions of the sodium salt of the ligand do barely emit, the two new SMOFs show a bright luminescence with a broad peak at about 440 nm. From experimental and computational results, it is quite evident that the metal atoms in the new compounds do not participate at all in the luminescence of the solid-state samples. In order to solve the apparent contradiction that intra-ligand transitions of a non-emitting ligand are at the origin of the luminescent emission in these compounds, we show via time dependent DFT calculations that coordination of a pair of anionic ligands around a central metal atom is essential to stabilize high energy lone pair orbitals in the carboxylate groups of the ligands, leading to a radical change of the nature of intra-ligand transitions for these ligands when incorporated into the ML_2_(H_2_O)_2_ units. This is, indeed, an indirect mechanism to induce emission upon aggregation that, to the best of our knowledge, has not been previously described in the literature. In this respect, we think that it is worth pursuing the search of H-bonded supramolecular organizations between metal complexes with this type of aggregation induced emission in the quest of new luminescent materials for solid state optoelectronic applications.

## Figures and Tables

**Figure 1 polymers-12-01756-f001:**
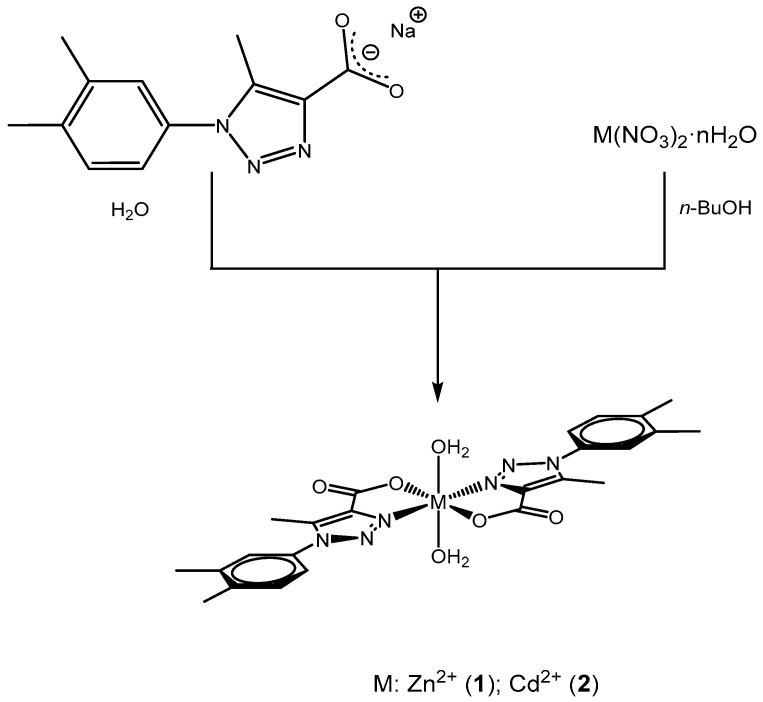
Scheme of the reaction to obtain complexes (**1**) and (**2**).

**Figure 2 polymers-12-01756-f002:**
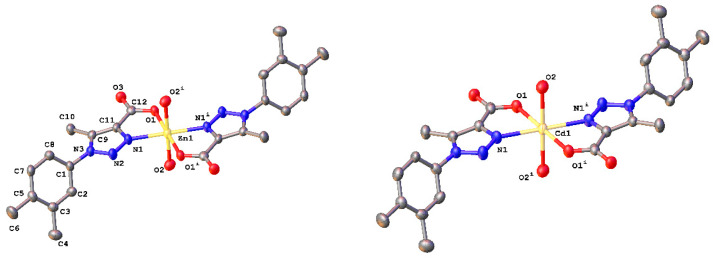
ORTEP plot for compounds **1** (left) and **2** (right). Hydrogen atoms were omitted for the sake of clarity. Thermal ellipsoids were drawn with a 30% of probability. Symmetry code: (i) $\bar{x},\;\bar{y},\bar{z}$.

**Figure 3 polymers-12-01756-f003:**
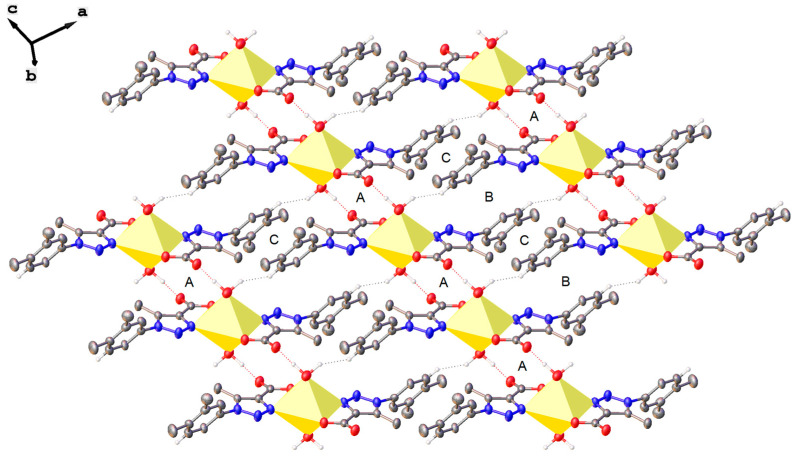
A view of the two-dimensional supramolecular aggregate in compounds **1** and **2** showing the formation of $R_{2}^{2}$(12), $R_{2}^{2}$ (26), and $R_{2}^{2}$ (24) rings (labeled A, B, and C, respectively). H atoms not directly involved in hydrogen bonds have been omitted for the sake of clarity.

**Figure 4 polymers-12-01756-f004:**
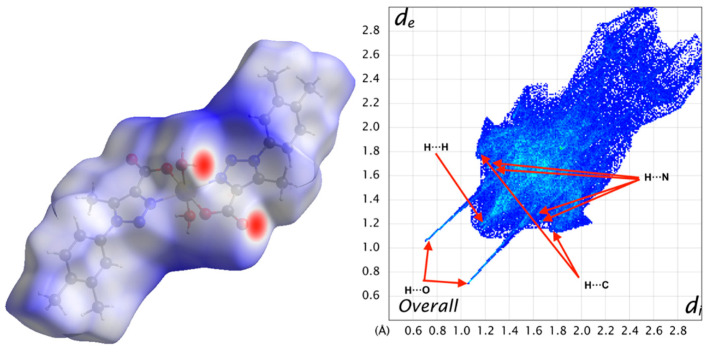
d_norm_ surface for the ZnL_2_(H_2_O)_2_ complex (left) and its 2D fingerprint plot (right). In the left panel, red and blue colors indicate strong and weak interactions, respectively. Isovalues range from −0.66 (blue) to +2.31 (red).

**Figure 5 polymers-12-01756-f005:**
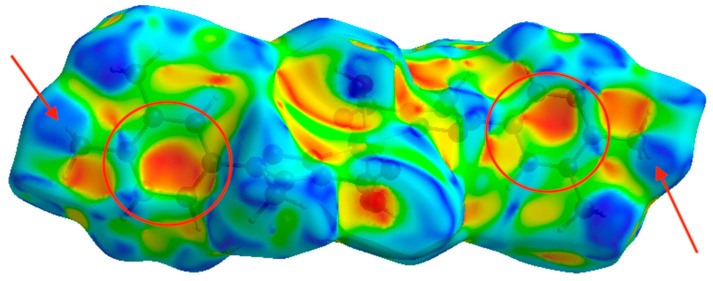
Shape index surface for ZnL_2_(H_2_O)_2_ complex. Red–orange and green–blue colors indicate zones engaging in weak interactions and their reciprocal contacts, respectively. Isovalues range from −1.0 (red–orange) to +1.0 (green–blue).

**Figure 6 polymers-12-01756-f006:**
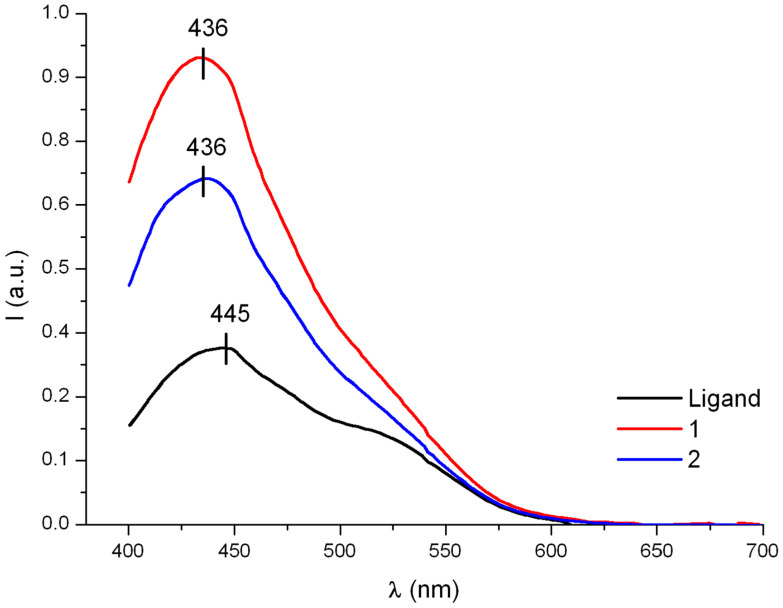
Emission spectra of the sodium salt of the ligand and the Zn^2+^ and Cd^2+^ complexes, all three in solid state.

**Figure 7 polymers-12-01756-f007:**
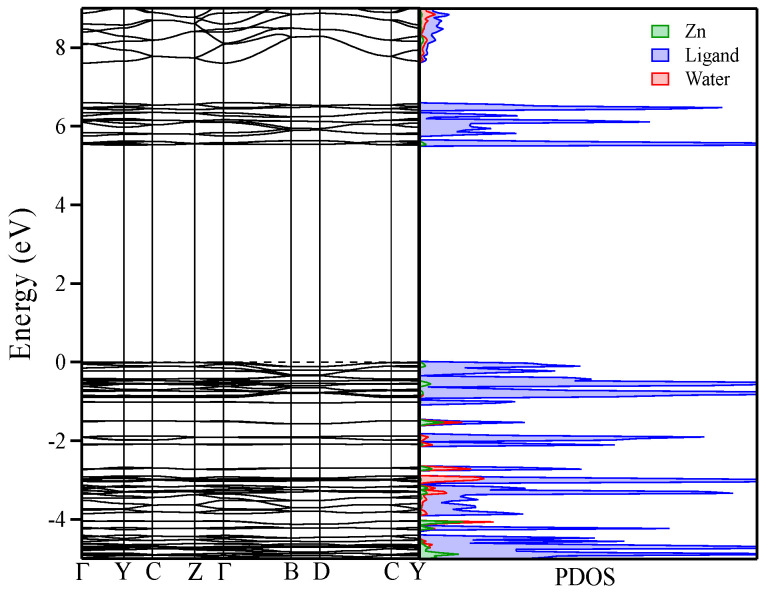
Band structure and partial density of states (PDOS) calculated with the B3LYP functional for compound **1** using the experimental X-ray diffraction structure.

**Figure 8 polymers-12-01756-f008:**
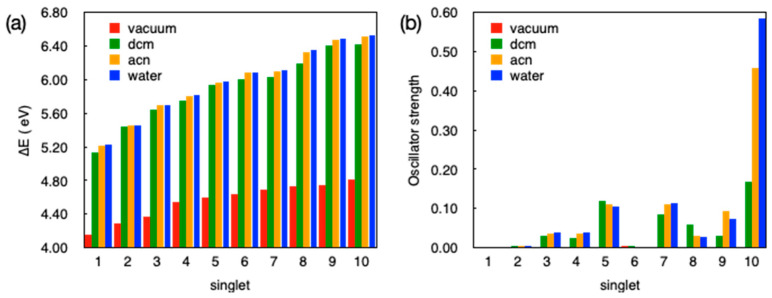
Vertical transition energies (**a**) and oscillator strengths (**b**) for the 10 lowest excited singlet states S_1_–S_10_ of the isolated ligand L^−^ in vacuum and in different solvents. The geometry of the ligand corresponds to that found experimentally for **1**.

**Figure 9 polymers-12-01756-f009:**
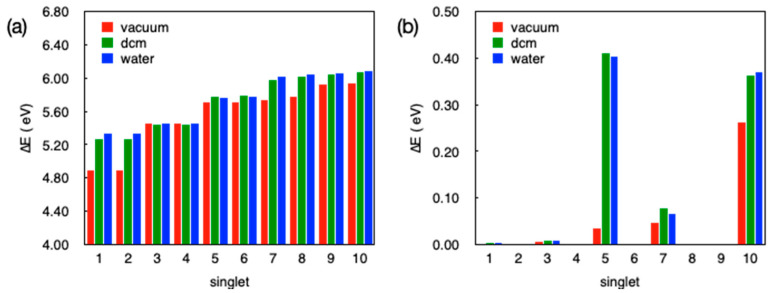
Transition energies (**a**) and oscillator strengths (**b**) for the ten lowest excited singlet states S_1_–S_10_ of the isolated ZnL_2_(H_2_O)_2_ complex in vacuum and in different solvents. The geometry of the complex considered in these calculations corresponds to the structural X-ray data obtained for **1**.

**Figure 10 polymers-12-01756-f010:**
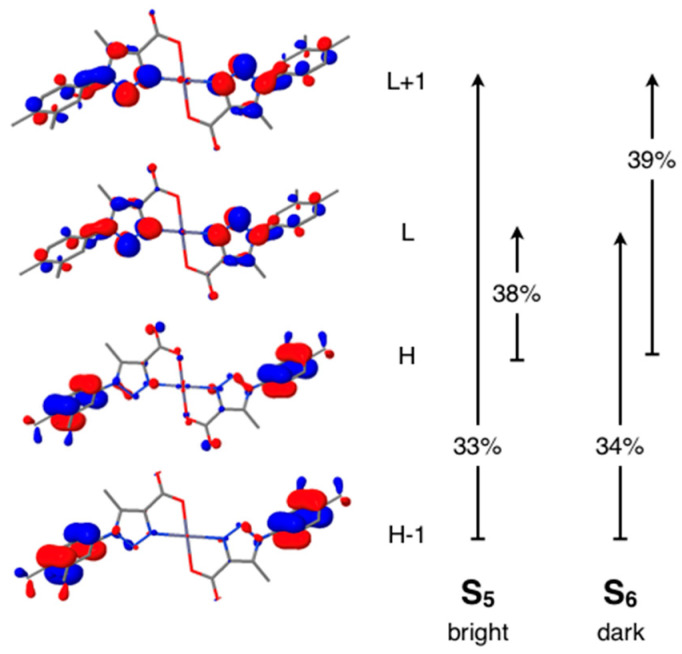
Frontier orbitals and main amplitudes for the S_5_ and S_6_ states for the ZnL_2_(H_2_O)_2_ fragment with the geometry obtained from the X-ray structure.

**Table 1 polymers-12-01756-t001:** Crystal data parameters for compounds **1** and **2**.

Compound	1	2
Empirical Formula	C_24_H_28_N_6_O_6_Zn	C_24_H_28_N_6_O_6_Cd
Formula mass, g mol^−1^	561.89	608.92
Collection T, K	295.48	295.62
Crystal system	Monoclinic	Monoclinic
Space group	*P2_1_/c*	*P2_1_/c*
*a* (Å)	11.6896(4)	11.8854(5)
*b* (Å)	16.7231(7)	16.8493(7)
*c* (Å)	7.2911(3)	7.4316(3)
*β* (°)	102.539(4)	105.219(3)
*V* (Å^3^)	1391.31(10)	1436.06(10)
*Z*	2	2
ρ_calcd_ (gcm^−3^)	1.341	1.408
Crystal size (mm)	0.227 × 0.087 × 0.050	0.099 × 0.077 × 0.068
*F* (000)	584.0	620.0
Abs coeff (mm^−1^)	1.628	6.481
*θ* range (°)	7.748/136.318	7.708/117.936
Range h,k,l	−14/12, −19/20, −8/8	−13/13, −18/18, −7/8
No. total refl.	12,232	12,407
No. unique refl.	2491	2057
Comp. *θ*_max_ (%)	98	100
Max/min transmission	0.923/0.709	0.577/0.643
Data/Restraints/Parameters	2491/0/174	2057/0/174
Final R [*I*>2*σ*(*I*)]	0.0686	0.056
R indices (all data)	0.1271	0.081
Goodness of fit/F^2^	1.029	1.085
Largest diff. Peak/hole (eÅ^−3^)	0.37/−0.34	0.82/−0.60

**Table 2 polymers-12-01756-t002:** Contributions of the principal intermolecular contacts (%) in compounds **1** and **2**.

Contact	1	2
H···H	45.2	41.5
H···C	16.3	15.7
H···N	9.9	10.0
H···O	25.3	24.7

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
