# Peer review of "Crystallization Induced Enhanced Emission in Two New Zn(II) and Cd(II) Supramolecular Coordination Complexes with the 1-(3,4-Dimethylphenyl)-5-Methyl-1H-1,2,3-Triazole-4-Carboxylate Ligand"

_polymers, 2020, doi:10.3390/polym12081756_

Round 1
Reviewer 1 Report
Manuscript by Alemany, Brito and co-authors reports on two new supramolecular metal-organic frameworks that show AIE at the solid state.
The work is well performed and presented.
Both SMOFs have bene structurally elucidated and Hirshfeld surface analysis were addressed.
Authors devoted several efforts in order to understand and explain photoluminescence origin with a wise use of periodic DFT and molecular TD-DFT calculations leading to consistent conclusions.
For these reasons, I suggest to accept the manuscript in its current form.
Author Response
We gratefully acknowledge the positive comments and the interest of this referee for our work.
Reviewer 2 Report
a) Manuscript has many typos, revise carefully and correct it
b) Manuscript needs to be revise by native English speaking
c) Improve 3.2. FT-IR spectroscopy, because explanation was poor and include spectra in the manuscript or in the answer to reviewers letter
d) Manuscript has figures, Figures, fig, and Fig, use Figures for all manuscript, and also change table for Table
e) Include important results in the Conclusion part
f) Manuscript has some interesting results, but discussion was very poor, improve discussion for all Figures
g) Manuscript has only one recently references from 2020 [References 78], two references from 2019 [31, 41]. Include more recently references specially from 2020
Author Response
- a) Manuscript has many typos, revise carefully and correct it
Answer: We thoroughly checked our manuscript and corrected several typos that we found in this revision.
- b) Manuscript needs to be revise by native English speaking
Answer: The manuscript has been revised and some changes were introduced according to this suggestion.
- c) Improve 3.2. FT-IR spectroscopy, because explanation was poor and include spectra in the manuscript or in the answer to reviewers letter
Answer: Many thanks for the comments. The discussion was revised and improved. The spectra were included in the Supplementary Information as Figure S1
- e) Include important results in the Conclusion part
Answer: The conclusions section was revised and we think that it now reflect clearly the most important results of our work.
- f) Manuscript has some interesting results, but discussion was very poor, improve discussion for all Figures
Answer: We thoroughly revised the manuscript trying to improve the discussion of our results.
- g) Manuscript has only one recently references from 2020 [References 78], two references from 2019 [31, 41]. Include more recently references specially from 2020
Answer: We appreciate the comment of the reviewer, but we feel that all references included in the manuscript are direct related with relevant results of the reported work.
Round 2
Reviewer 2 Report
I didn’t agree with answer
------------------------------------------
g) Manuscript has only one recently references from 2020 [References 78], two references from 2019 [31, 41]. Include more recently references specially from 2020
Answer: We appreciate the comment of the reviewer, but we feel that all references included in the manuscript are direct related with relevant results of the reported work
------------------------------------------
If only is important previous publications, why publish new results. Authors didn’t want to correct this part with superficial answer